# The Impact of Scholastic Factors on Physical Activity Levels during the COVID-19 Lockdown: A Prospective Study on Adolescents from Bosnia and Herzegovina

**DOI:** 10.3390/children8100877

**Published:** 2021-10-01

**Authors:** Damir Sekulic, Daria Ostojic, Andrew Decelis, José Castro-Piñero, Tatjana Jezdimirovic, Patrik Drid, Ljerka Ostojic, Barbara Gilic

**Affiliations:** 1Faculty of Kinesiology, University of Split, 21000 Split, Croatia; dado@kifst.hr (D.S.); ljerka.ostojic@mef.sum.ba (L.O.); 2School of Medicine, University of Mostar, 88000 Mostar, Bosnia and Herzegovina; dariaostojic@gmail.com; 3Faculty of Education, Institute for Physical Education and Sport, University of Malta, MSD 2080 Msida, Malta; andrew.decelis@um.edu.mt; 4GALENO Research Group, Department of Physical Education, Faculty of Education Sciences, University of Cádiz, Avenida República Saharaui s/n, 11519 Puerto Real, Spain; jose.castro@uca.es; 5Instituto de Investigación e Innovación Biomédica de Cádiz (INiBICA), 11009 Cádiz, Spain; 6Faculty of Sport and Physical Education, University of Novi Sad, 21102 Novi Sad, Serbia; tatjanaj.ns@gmail.com (T.J.); patrikdrid@gmail.com (P.D.); 7Academy of Medical Sciences of Bosnia and Herzegovina, 71000 Sarajevo, Bosnia and Herzegovina; 8Faculty of Kinesiology, University of Zagreb, 10000 Zagreb, Croatia

**Keywords:** physical activity, puberty, pandemic, health literacy, academic achievement

## Abstract

Scholastic factors (academic achievement) are hypothesized to be important determinants of health-related behaviors in adolescents, but there is a lack of knowledge on their influence on physical activity levels (PAL), especially considering the COVID-19 pandemic and the imposed lockdown. This study aimed to investigate the associations between scholastic factors and PAL before and during the pandemic lockdown. The participants were adolescents form Bosnia and Herzegovina (*n* = 525, 46% females), who were observed prospectively at the baseline (before the pandemic lockdown) and during the lockdown in 2020 (follow-up). The scholastic factors (grade point average, behavioral grade, school absences, unexcused absences) were evidenced at the baseline (predictors). The outcome (PAL) was evaluated using the Physical Activity Questionnaire for Adolescents at the baseline and the follow-up. Gender, age, parental/familial conflict, and sport participation were observed as confounders. No significant influence of the predictors on PAL were evidenced at the baseline or at the follow-up. The scholastic variables were significantly associated with the changes of PAL which occurred due to pandemic lockdown, with a lower risk for negative changes in PAL among adolescents who were better in school (OR = 0.56, 95%CI: 0.34–0.81, and OR = 0.66, 95%CI: 0.34–0.97, for the grade point average and behavioral grade, respectively). Students who do well in school are probably more aware of the health benefits of proper PAL, and therefore are devoted to the maintenance of their PAL even during the home-confinement of lockdown. Public health authorities should focus more on helping adolescents to understand the importance and benefits of proper PAL throughout the school system.

## 1. Introduction

There is a global trend of insufficient physical activity (PA) in adolescents, putting their current and future health at risk [1,2]. Specifically, 81% of adolescents are reported to be insufficiently physically active worldwide, meaning that they do not reach the WHO’s recommendation of 60 min of PA a day [1]. In support of this, a recent paper reported that 56% of girls and 44% of boys in Bosnia and Herzegovina perform insufficient PA [3]. It is well known that sufficient PA decreases the incidence of various chronic diseases, including metabolic, respiratory and cardiac diseases, and numerous forms of cancer [4]. Thus, it is of great importance to promote PA during adolescence, as this is a critical period for establishing behavior patterns and personal lifestyle choices [5]. What is more, there is firm evidence that health habits during adolescence track into adulthood, with low PA tracking better than high PA [6]. A Finnish study found that adolescents with low PA had a higher probability of being smokers in adulthood, which is an additional detriment to health [7]. Not surprisingly, numerous studies have focused on the identification of the determinants and factors that affect PA in adolescents, in order to develop specific strategies for PA promotion [8,9,10].

The most common and most investigated factors of influence on PA in adolescents are demographic (age, gender), social (family and friend support), behavioral (participation in sports, substance abuse), environmental (social, built, and natural environment), and psychosocial (self-efficacy, motivation, perceived competence, the confidence to be physically active) [9,11,12]. In general, adolescents’ PA is positively associated with male gender, perceived activity competence, intentions to exercise, previous childhood PA, social support, family cohesion, the families’ high socioeconomic status, and opportunities to exercise [13]. Conversely, an inverse relationship with PA has been recorded for increased age, smoking, unhealthy diet, depression, and sedentary behavior after school and on weekends [9]. Interestingly, studies have evidenced a positive association between sedentary activities related to school (i.e., doing homework, reading, studying) and PA, indicating that specific correlations may exist between scholastic factors/academic achievement and PA in adolescence [14,15].

For example, academic achievement is positively associated with higher physical activity levels (PAL) in adolescents from the USA, meaning that adolescents who have better grades in school tend to be more physically active [16]. Similarly, higher school grades predict higher PAL in adolescents from the USA [15]. While there is certain evidence that PA positively influences cognitive capacities [17,18], it is also theorized that adolescents with better academic achievement and better school behavior are more likely to assume positive health behaviors, including the importance of being physically active. Indeed, adolescents who spend time improving their academic performance and are more productive during their leisure time are more likely to spend time doing PA [14]. Adolescents who productively spend their after-school time doing homework and studying have better time-management skills and, thus, can better create time for PA. On the other hand, individuals who spend time engaging in other sedentary behaviors (e.g., watching TV, playing video games) are reported to be less physically active and more involved in health-risk behaviors [14,19]. However, the associations between academic achievement and PAL are not conclusive; a study on Chinese adolescents has evidenced no association [20], while a study on Korean adolescents evidenced a negative association [21] between these factors. Therefore, the problem should be further examined, and special emphasis should be placed on the possible cultural differences that may have caused such inconclusive results. 

The COVID-19 pandemic was declared in March 2020, and since then has affected the lives of people all around the globe. Numerous strategies for reducing the incidence of infection have been implemented, with social distancing measures being the most notable. Most places where many people can gather (e.g., schools, universities, sports facilities) were closed [22]. Therefore, as movement opportunities decreased, studies consistently reported a decrease in PALs among adolescents globally [23,24,25], including in Southeastern Europe [26,27]. Not surprisingly, studies investigating PAL and changes in PAL during the COVID-19 pandemic have also aimed to determine the factors that influence PA, through comparison to investigations undertaken in “regular” circumstances.

In brief, it has been shown that boys’ PAL declined more than the PAL of girls due to the higher involvement of boys in organized sports activities in the pre-pandemic period, as these activities were heavily restricted during the pandemic [28,29,30]. In support of this, active adolescents decreased the intensity of the PA more than their less-active peers [31]. Furthermore, environmental factors influenced PAL significantly, as adolescents from urban environmental settings decreased their PAL to a greater extent compared to their rural peers [29,32]. This was related to the higher access to sports facilities among urban adolescents in normal circumstances, and the closure of those facilities during the pandemic. Conflict with parents was negatively associated with PA, as parents who were in conflict with their children were not able to efficiently encourage their children to be physically active during the pandemic [24,28]. Moreover, a recent study evidenced that pre-pandemic sports participation and fitness status was positively associated with PALs during the pandemic, which was explained by the better physical literacy of children involved in sports, and their motivation to stay physically active even in “irregular” circumstances, such as COVID-19-imposed lockdowns [28].

From the previous literature overview, it is evident that studies have found various factors that influence PAL in adolescents during the COVID-19 pandemic (i.e., gender, environment, social, and sports factors). However, no study so far has documented the influence of scholastic factors on PAL in this challenging situation. Therefore, this study aimed to investigate the influence of scholastic factors (i.e., academic achievement, behavioral grade, school absences) on PAL in adolescents from Bosnia and Herzegovina (BH) before and during the COVID-19 lockdown, and to evaluate their possible influence on changes in PAL which occurred as a result of the imposed measures of the lockdown due to the COVID-19 pandemic. We hypothesized that students with better academic achievement generally act more responsibly towards themselves and their success. Furthermore, we theorized that they will act more responsibly towards their health, they will have the knowledge and awareness of the importance of PA for their health, and that scholastic factors would be protective against PA decline in adolescents. We were of the opinion that the results of this study would be applicable to the development of targeted campaigns against the known and widespread decrease of PAL in adolescence, not only in the situation of the COVID-19 pandemic, but also during other health crises, weather crises, environmental crises, or other situations where there may be limited movement opportunities.

## 2. Materials and Methods

### 2.1. Participants and Study Design

This prospective study included total of 525 adolescents from Bosnia and Herzegovina (43% females). At the study baseline, the participants were 16.2 ± 2.3 years of age. Before the COVID-19 pandemic, all of the participants regularly participated in physical education (PE) classes and were in good health (i.e., were not ill or injured). This study was part of other previously initiated research (“physical activity, substance misuse, and factors of influence in adolescence”) approved by the Ethical Board of the University of Split, Faculty of Kinesiology (EBO: 2181-205-05-02-05-20-004). Therefore, all of the participants were previously informed about the study aims and benefits, and the parents signed informed consent forms before the study baseline.

The study included two measurement points (testing waves): (1) the baseline, conducted before the lockdown period; and (2) the follow-up, undertaken during the lockdown period under strict social distancing due to the COVID-19 pandemic. The baseline measurement included the assessment of PAL before the lockdown, scholastic factors, and demographic variables. The follow-up measurement included the assessment of PALs during the lockdown. During the lockdown, schools were completely closed, and classes were held online. Regarding PE, it was also conducted online. There was no uniformed direction of online PE, but each teacher made classes as they wanted and were able to, according to their technical equipment and organizing skills. Mostly, PE was held in the form of encouraging students to be active in their home or safe open spaces, with information about proper exercise form. For a more detailed description of the study design, please see Figure 1.

### 2.2. Variables and Measurement

The variables included sociodemographic characteristics (gender and age), physical activity levels (PAL), and scholastic factors. Additionally, as previous studies had shown the significant influence of certain variables on PALs during the COVID-19 lockdown, we considered sports participation, parental education, and parental/familial conflict, although those variables were not directly related to the study aims (for more details, please see the Introduction). 

In order to assess PAL, the adolescents filled in the online form of the Physical Activity Questionnaire for Adolescents (PAQ-A). The adolescents filled the PAQ-A on two occasions: first, before the lockdown period (baseline PAL), and second, during the lockdown period (follow-up PAL). The PAQ-A is a self-administered questionnaire designed for adolescents from 14 to 19 years old, which includes questions regarding PA during the last 7 days [33]. The PAQ-A consists of nine items assessing the frequency of participation in different types of PA (i.e., PA during physical education classes, school recess, free play, sports). The results of each item and the total score are scaled from 1 to 5, representing low to high PAL, respectively [34]. In this study, we observed the crude results of PAL at the baseline (PALBL), and PAL at the follow-up (PALFU). Next, the crude PAL was also observed as a binomial variable with two categories: results lower than 2.73 were classified as insufficient PAL, and results higher than 2.73 were marked as sufficient/normal PAL, as previously suggested [34]. Further, to quantify the changes in PALBL and PALFU, we calculated the crude numerical difference between these two values (PALΔ = PALBL − PALFU). Next, we calculated the relative changes in PAL between the baseline and follow-up (in %) using the following calculation: PALΔ% = (PALBL − PALFU)/PALBL × 100. For the purpose of later statistical calculations, the participants were ordered according to their PALΔ%, and then grouped into two groups (0–50th percentile, and above). The participants with a greater relative decline of PAL (in the above-50th percentile group) were the “high-risk group”, while those in the first 50 percentiles were the “low-risk group”. Such dichotomization allowed us to calculate the logistic regression for PALΔ% as a binomial criterion. 

Scholastic factors included academic achievement (grade point average, GPA), school absences, and behavioral grade. The participants were asked about their GPA, representing their academic achievement over last semester, assessed on a five-point scale ranging from 1 to 5, representing excellent to poor achievement. Unexcused school absence was the number of unexcused absences in school hours in one year, presented on a five-point scale covering <5 h, 5–10 h, 11–15 h, 16–20 h, and >20 h. Overall school absence was assessed on a four-point scale: almost never, rarely, from time to time, and often. Behavioral grades were evaluated on a five-point scale ranging from 1 to 5, representing poor to excellent behavior. All of the scholastic variables had been used in previous studies for similar participants, and had shown appropriate reliability and validity [35,36].

Sports participation was evaluated based on years of sports involvement, including the following answers: never involved, less than one year, 2 to 5 years, and more than five years [35]. Parental education was determined based on the maximum education level/degree, including the following answers: elementary school, high school, college degree, and university degree. Parental/familial conflict was assessed with the question “How often do you have a conflict with your parents/family?”, with the following possible answers: never, rarely, from time to time, and regularly [26].

### 2.3. Statistical Analyses

The normality of the distribution was checked using Kolmogorov–Smirnov’s test, and the descriptive statistics included means and standard deviations (for numerical variables), and frequencies and percentages (for ordinal and nominal variables). 

The differences between the groups were evidenced by the Mann–Whitney Z test (MW) (for ordinal variables), and Chi square (χ^2^). Spearman’s rank order correlation was calculated in order to evidence the associations between age and PAL at the baseline and follow-up. The T-test for dependent samples was used to identify the changes in PAL between the baseline (pre-pandemic period) and follow-up (lockdown period). 

Logistic regression (with an Odds Ratio (OR) and a 95% Confidence Interval (95%CI) presented) was applied to show the association between the predictors and the categorized PAL (insufficient PAL, coded as “1” vs. sufficient PAL, coded as “2”) at the baseline and follow-up. Additionally, in order to identify any existing association between the predictors and changes that occurred in PAL, we calculated the logistic regression with dichotomized PALΔ% as a criterion (low-risk group, coded as “1”, vs. high-risk group, coded as “2”). Because preliminary analyses showed significant associations between age, sport participation, parental/familial conflict, parental education and gender with PAL (please see the Results Section for more details), the logistic regressions were controlled for confounders (age, sport participation, male gender, parental education, and parental/familial conflict). 

A *p*-Value of 95% was applied, and the Statistica ver. 13.5 statistical package (Tibco Inc., Palo Alto, CA, USA was used for all of the calculations.

## 3. Results

Descriptive statistics (frequencies and percentages) of the studied variables for the total sample are presented in Appendix A. 

The PAL significantly declined between the baseline and follow-up (2.43 ± 0.71 and 2.00 ± 0.75, respectively; *t*-test = 4.14, *p* < 0.001), indicating a negative impact of the COVID-19-imposed lockdown on PAL among the studied adolescents. 

The differences between the groups of adolescents based on sufficient/insufficient PAL at the baseline are presented in Table 1. The sufficient PAL was more prevalent among boys (χ^2^ = 70.01, *p* < 0.01), those adolescents who were involved in sports (MW = 7.89, *p* < 0.001), adolescents whose parents were better educated (MW = 3.69, *p* < 0.001), and those who reported a lower level of conflict with their parents/family (MW = 4.64, *p* < 0.01). Scholastic factors did not differentiate the groups clustered according to PAL sufficiency/insufficiency at the baseline.

Table 2 presents the differences between those adolescents who achieved sufficient and those who had insufficient PAL at the follow-up (during lockdown period). Again, boys were more likely to have sufficient PAL then girls (χ^2^ = 53.78, *p* < 0.01). Furthermore, sufficient PAL was more prevalent in the adolescents who, at the baseline, reported higher involvement in sports (MW = 6.67, *p* < 0.001), those whose parents were better educated (MW = 2.55, *p* < 0.01), and those who reported a lower level of conflict with parents/family members (MW = 3.4, *p* < 0.01). 

The results of the logistic regression analyses for dichotomized outcomes (PAL at the baseline, PAL at follow-up, changes in PAL due to COVID-19 lockdown) are presented in Figure 2. The logistic regression calculated at the baseline indicated no significant association between the scholastic variables and sufficient PAL before the pandemic lockdown. Scholastic variables were not significantly associated with sufficient PAL during lockdown. GPA and the behavioral grade observed at the baseline were significantly associated with dichotomized PALΔ%. In brief, a lower likelihood of being in the high-risk group for the decline of PAL due to the pandemic lockdown was evidenced in adolescents who had better grades in school (OR = 0.56, 95%CI: 0.34–0.81, and OR = 0.66, 95%CI: 0.34–0.97, for GPA and behavioral grade, respectively). 

## 4. Discussion

The aim of this study was to investigate the associations between scholastic factors and PAL before and during the COVID-19 lockdown, and to evaluate its influence on changes in PAL due to the COVID-19 lockdown. The results showed no associations between scholastic factors and PALs before and during the lockdown period. However, scholastic factors (GPA and behavioral grade) were associated with the changes of PAL that occurred due to the lockdown. 

### 4.1. GPA and PAL before and during the COVID-19 Lockdown

An association between GPA and PAL, both before and during the lockdown, was not found. However, this was not unexpected, as previous studies on similar samples of high-school students were inconsistent with regards to associations between academic achievement and PAL. Studies on Canadian, American, and Norwegian high-school students found that adolescents with higher school grades had higher PALs [14,16,37]. The authors explained their results by assuming that individuals who are more committed to school duties have better self-discipline and time management skills, and consequently make more time for PA. Additionally, intelligent students with good academic achievement were thought to need less time for studying, and thus have more time for PA [38]. On the contrary, a study on adolescents from Korea showed a negative association between academic achievement and PAL [21], which was supported by Coleman [39], who argued that this negative association is due to the belief that high PA drains children’s energy and has a debilitating effect on academic attainment. Similar to our study, a study on high-school students from England found no associations between school grades and PAL [40]. There could be several possible explanations, such as cultural differences and environmental situations, but the authors of that study explained such findings by the type of school. They investigated students from one “good” school that enrolls mainly good students; thus, academic grades clustered towards the higher end, restricting the variability within the academic scores. It can be assumed that the type of school could have affected the associations in our study as well.

The lack of association between GPA and PAL in our study can be explained by the fact that academic achievement in high school largely depends on the school quality, type, size, and location [41]. Our investigation included adolescents from different schools that are known to vary in the demands of their academic program (unlike the previously discussed English study of Daley and Ryan [40]). This means that children with average academic achievement in a high-quality school would most likely have excellent achievement if they were students in a less demanding school program. It implies that academic achievement expressed by GPA cannot be objectively compared between different schools. As a result, the previously established explanation for the association between academic success and PAL (i.e., more intelligent children need less time for academic duties and have more free time for being physically active) cannot be applied in our investigation. As it was noted that students from different schools are not comparable based on grades, the Matura exam at the end of high school education was implemented in the country to improve the objectivity of academic achievement [42]. However, in our study, we used GPA, as we included students from first to fourth grade who had not completed the Matura exam yet. 

### 4.2. Behavioral Grade and PAL before and during the COVID-19 Lockdown

Our study is among the first in which the behavioral grade was correlated with PAL in adolescents, and as far as we are aware, probably the first one where this issue was studied in the COVID-19 period. Behavioral grades have so far been associated with health-risk behaviors, most commonly with substance use and misuse (SUM). Thus, in previous studies on similar samples of students from southeastern Europe, it was noted that behavior grades are the strongest predictor of SUM (smoking, alcohol, drugs) [35,36]. Specifically, studies have noted that adolescents with poor behavior grades have a higher risk of engaging in health-risk behaviors [35,43]. Therefore, we initially assumed that there would be associations between behavior grades and PAL (as a specific type of health-risk behavior); however, this was not the case. The most likely reason for this is that behavior grades are given based on subjective evaluations and perceptions of students’ behavior [20].

The behavior grade indicates socially undesirable and problematic behaviors that teachers manage to recognize and, based on that, give a bad behavior grade. Therefore, children with poor behavior grades are often those children who partake in health-risk behavior (such as SUM, especially smoking). This can be explained by the theory that health-risk behaviors often occur together; that is, one behavior often leads to another [44]. Related to this, adolescents who are not good at school may be in non-controlled social environments, and may be more socially influenced by peers involved in undesirable and problematic behaviors [35]. At first glance, it seems that there should also be a link between poor behavior grades and low PAL (which is also a health-threatening behavior). However, low PAL, cannot be considered an undesirable behavior, because it can be caused by some “objective” factors such as obesity, socioeconomic status, depression, screen time, and other sedentary behaviors, which are generally not related to a non-controlled social environment, which is regular determinant of SUM [45]. This most likely explains the lack of association between behavioral grade and PAL, irrespective of previous findings where a poor behavioral grade was associated with other health-threatening behaviors in adolescents.

### 4.3. Absences from School and PAL before and during the COVID-19 Lockdown

The association between absence from school and PAL before and during the pandemic and PAL changes due to lockdown was not recorded, due to the confounding influence of sports participation. In brief, both adolescents who participate in sports and adolescents who deliberately skip school have a high number of absences. As sport makes up a large part of the total PAL, adolescents involved in sports activities (especially competitive and higher-level sport) tend to have a high PAL [46]. This was confirmed even in our study (please see Results). However, adolescents who participate in sports generally have more absences from school due to competitions, training camps, and other sport-related travels. Therefore, adolescents involved in sports regularly have a high number of absences from school due to sport-related obligations, and not due to intentional and unexcused skipping school [47].

On the other hand, another group of adolescents who are not involved in sports also have a high number of school absences, but at the same time, most likely have low PAL. Outside of absenteeism related to out-of-school duties (i.e., sports participation), school absenteeism is considered a predictor of future risk behaviors, including delinquency, school dropout, and SUM [48]. Additionally, absenteeism has been associated with adolescents distancing themselves from school-based health programs, exposing them to increased chances for indulging in health-risk behaviors [49]. Therefore, adolescents with a high number of absences from school are commonly characterized as deviant or risk-taking. Such adolescents usually do not take proper care of their health and have a lower PAL [50]. 

Putting this all together, there is conclusive evidence that adolescents who deliberately skip school most likely have low PAL, while adolescents involved in sports have a high PAL [51]. However, both “groups” of adolescents probably have a similar number of absences from school. This altogether most likely resulted in the lack of correlation between school absenteeism and PAL in this study.

### 4.4. Scholastic Factors and Changes in PAL Due to the COVID-19 Lockdown

One of the most important findings of this study is that adolescents with better academic achievement were less likely to have a significant drop in PAL due to the COVID-19 lockdown. While this is, to the best of our knowledge, one of the first studies where this issue has been explained in the context of the COVID-19 pandemic (or a similar crisis), we will first give an overview of the previous studies where similar issues were examined under ordinary circumstances. 

In brief, a longitudinal study in Finland that investigated individuals first at 14 and then at 31 years of age found that poor academic achievement in adolescence was associated with a low PAL in adulthood [52]. Similarly, higher childhood educational levels were associated with more frequent participation in sports and recreational activities in adulthood in individuals from the United Kingdom [53]. Furthermore, a few studies found that adolescents with a better GPA had a higher PAL, which has mostly been explained by the assumption that those adolescents have better time management skills, make better lifestyle choices, and generally behave more responsibly [14,37]. 

Although the previously cited studies are not directly related to changes in PAL, their findings are important for our investigation because they at least partially contextualized the influence of scholastic achievement on the PAL changes that occurred during the lockdown in our study. In brief, as schools were closed and adolescents shifted to attending online school classes during the COVID-19 lockdown, they had to create new daily routines, including attending online classes, studying, and involvement in other non-school activities (e.g., exercise and other forms of PA). As adolescents who perform better in school under normal circumstances tend to manage their life more efficiently [14,37], it could be expected that they would also be capable of doing so even during the pandemic period. Furthermore, those students might possess better physical literacy, which gives them confidence and self-esteem for being physically active even in a different space and with limited equipment for practicing sports or exercising [54].

With regards to PA, it is important to note that adolescents with a better GPA (better educated) also possess better health literacy (HL). In the most general terms, HL can be defined as “cognitive and social skills, which determine the motivation and ability of individuals to gain access to, understand, and use information in ways, which promote and maintain good health” [55]. Indeed, the studies performed so far have demonstrated positive correlations between educational level and higher HL in Australia and Denmark [56,57]. Furthermore, HL affects students’ understanding of information regarding the significance of health, and the ability to make proper health decisions [58]. It was also noted that students with higher HL tend to adopt numerous behaviors which are beneficial to their health, including PA [59], which has additionally been supported by studies showing that adolescents with higher levels of HL have higher PALs [60,61].

Furthermore, studies carried out previously have already noted better HL in adolescents who do better in school [57,61]. For example, a Finnish study found that adolescents with higher academic achievement had better HL, and more frequently participated in sports activities [61]. Similarly, it was noted that HL mediates the association between educational achievement and PA in older participants from Denmark [57]. Hence, it is reasonable to assume that adolescents with higher school achievement can better understand the importance of PA for their health, and therefore may strive to be sufficiently physically active. It is possible, then, that the adolescents with higher academic achievement in our study possessed better HL skills and recognized the importance of maintaining their PAL during the pandemic in order to preserve their health. Only adolescents with adequate HL skills are likely to be able to find, understand, and use the information regarding the influence of PA on their health. Thus, it was expected that adolescents with better HL would be more likely to strive to maintain their PAL during the pandemic, based on the information on the positive influence of PA on the immune system (i.e., PA boosts the immune system and improves the cardiovascular and respiratory system, which is important for fighting against COVID-19 and preserving health in general) [62]. The stated explanation was actually supported by recent studies, in which physical literacy was found to be a factor of significant influence on PAL during lockdown [28]. Given the similarity between HL and physical literacy [63], the positive influence of HL on PAL (and GPA as a measure of overall educational success) is logical. 

It must be noted that not only GPA was associated with PAL changes; a lower decrease in PAL was observed even in adolescents with better behavioral grades. As has already been stated, previous studies suggested that behavioral grades are mainly related to health-risk behaviors (e.g., SUM) and youth who are not striving for academic achievement [35]. Logically, if we assume that adolescents with poor behavior grades are prone to health-risk behaviors and do not take care of their health [43], it could be expected that those adolescents would not make an effort to maintain regular PAL during the lockdown (irrespective of their PAL before pandemic). Further, it was noted that students with higher academic achievement attain better school behavioral grades [20]. Therefore, it could be assumed that adolescents with higher behavior grades will also have better knowledge of the importance of maintaining PAL during the lockdown. Likewise, our finding that adolescents with better school behavior succeeded in maintaining PAL during the pandemic could be explained by more pronounced responsible behavior, in this case towards themselves and their health, irrespective of the unfriendly conditions and environment children experienced during the lockdown.

One could argue that if such correlations between scholastic variables and PAL-changes truly exist, then they would be expected to appear even when the GPA and behavioral grade were correlated to PAL (and not PAL changes), which was not the case (note that scholastic variables were not significantly associated with PAL before and during lockdown). However, it must be noted that PAL can be influenced by numerous factors (i.e., opportunities to be active, living environment, sport preference). On the other hand, the changes in PAL which occurred due to the lockdown actually represent “changes in health-behaviors”, and therefore “responsible life behaviors” are more likely to be associated with them.

### 4.5. Limitations and Strengths

The main limitation of this study is the self-reported assessment of PAL, and the possibility of self-report bias. However, the participants filled out the questionnaire anonymously (e.g., they used self-selected codes for the tracking of responses), and identical procedures were used for the baseline and follow-up assessment, which we believe mostly eliminated this issue. Additionally, the used questionnaire for PA assessment did not evaluate the intensity of PA, which could display in more detail the situation with changes in PA. Furthermore, this study used the complete PAQ-A questionnaire, with the possibility that some questions are not applicable for the lockdown period. However, we did include the full questionnaire, as we initially collected all of the data and continued in the same way during the lockdown. In this research, the PAQ-A results were clustered into two groups, which undoubtedly allocated participants with similar results into opposed groups. In future studies, division into three groups is therefore suggested. This study emphasized the problem of differences in “school quality” in the analysis of the associations between educational achievement and different criteria, which limited the ability to discuss the obtained results in this context. Finally, this study included adolescents from only one country; future studies should evaluate adolescents from different countries in order to obtain more detailed results. 

To the best of our knowledge, this study is among the first to investigate the association between scholastic factors and PAL during the COVID-19 lockdown, which is the most important strength of this research. Furthermore, this study used the same assessment tools as previous studies on similar topics, making the results comparable. This investigation made possible future investigations of the factors affecting PAL during pandemics and similar crisis situations, which is important given that, overall, PALs in adolescents are alarmingly insufficient, especially considering the negative influence of the COVID-19 pandemic. 

## 5. Conclusions

The results showed no associations between scholastic factors and the PAL of adolescents before and during the COVID-19 lockdown. The main reason for the lack of association between GPA and PAL could be that academic achievement in high school largely depends on the school quality. Furthermore, the lack of association between behavioral grades and PAL could be attributed to the fact that a behavioral grade mostly represents teachers’ perceptions of their students, which is based on risk-taking behaviors (i.e., smoking and drinking) but rarely includes low PA as a significant health-threatening behavior. Finally, the fact that students who participate in sports and who intentionally skip school both have a high number of absences probably explains the lack of correlation between school absences and PAL in our study.

The study’s most important finding is that GPA and behavioral grades were associated with PAL changes during the lockdown, implying that adolescents with better scholastic attainment were less likely to reduce their PAL due to unfavorable conditions during the lockdown. It is likely that students who do well in school (irrespective of the quality of the school) are more aware of the health benefits of PA (i.e., they have better health literacy skills), and will therefore be more likely to try to maintain their PAL even during the home-confinement of lockdown, but this should be further investigated in future investigations.

Collectively, our results suggest that public-health policies should be more oriented towards improving HL in adolescents; that is, helping them to understand the importance and benefits of PA for their health, in order to improve and increase their PAL. Future studies should directly and in more detail evaluate the relationships among scholastic factors, HL, and PAL, in order to enable public health authorities to create more specific programs for the improvement of both HL and PAL in adolescents.

## Figures and Tables

**Figure 1 children-08-00877-f001:**
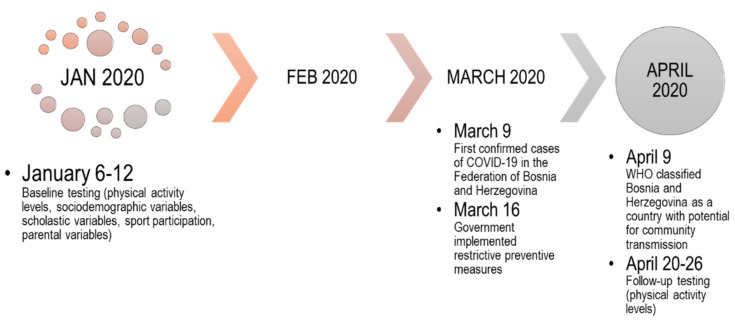
Study protocol and the most important dates.

**Figure 2 children-08-00877-f002:**
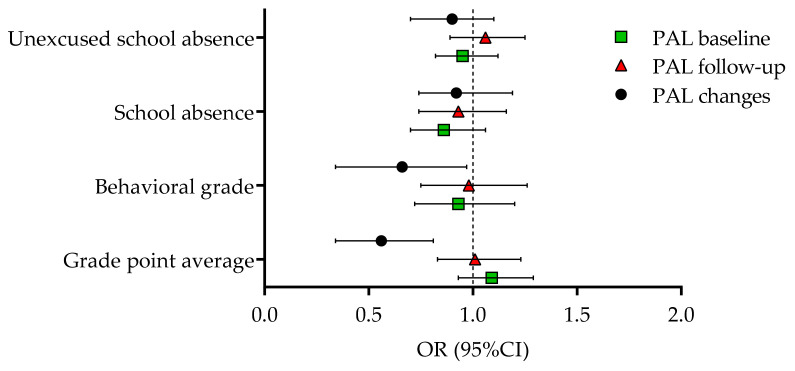
Logistic regression results for the dichotomized criteria: PAL at the baseline, PAL at follow-up (during lockdown), and PAL changes between the baseline and follow-up.

**Table 1 children-08-00877-t001:** Descriptive statistics (F—frequencies; %—percentages) for the study variables, with differences between groups according to the insufficiency/sufficiency of the physical activity levels (PAL) at the baseline (MW—Mann–Whitney test; χ^2^—Chi square test).

	Insufficient PAL		Sufficient PAL		MW/χ^2^	
	F	%	F	%	Z/χ^2^	*p*
Gender ^χ2^						
Male	142	40.00	142	78.89		
Female	208	58.59	38	21.11		
Missing	5	1.41	0	0.00	70.01	0.001
Grade point average						
Excellent (5)	142	40.00	63	35.00		
Very good (4)	133	37.46	67	37.22		
Good (3)	62	17.46	42	23.33		
Sufficient (2)	5	1.41	2	1.11		
Insufficient (1)	13	3.66	4	2.22		
Missing	0	0.00	2	1.11	0.89	0.37
School absences						
<5 h (4)	146	41.13	83	46.11		
5–10 h (3)	148	41.69	60	33.33		
11–20 h (2)	52	14.65	28	15.56		
>20 h (1)	9	2.54	9	5.00		
Missing	0	0.00	0	0.00	0.36	0.71
Unexcused absences	Count	Percent	Count	Percent		
<5 h (5)	253	71.27	127	70.56		
6–10 h (4)	66	18.59	32	17.78		
11–15 h (3)	19	5.35	5	2.78		
16–20 h (2)	8	2.25	7	3.89		
>20 h (1)	9	2.54	9	5.00		
Missing	0	0.00	0	0.00	0.34	0.73
Behavioral grade						
Excellent (4)	304	85.63	152	84.44		
Very good (3)	39	10.99	13	7.22		
Proper (2)	5	1.41	11	6.11		
Poor (1)	7	1.97	4	2.22		
Missing	0	0.00	0	0.00	0.54	0.58
Sport participation						
Never been involved	125	35.21	20	11.11		
<1 year	79	22.25	28	15.56		
2–5 years	97	27.32	57	31.67		
>5 years	54	15.21	75	41.67		
Missing	0	0.00	0	0.00	7.89	0.001
Parental education						
Elementary	33	9.30	7	3.89		
High school	265	74.65	123	68.33		
College degree	33	9.30	27	15.00		
University degree	24	6.76	23	12.78		
Missing	0	0.00	0	0.00	3.69	0.001
Parental conflict						
Never	119	33.52	94	52.22		
Rarely	142	40.00	62	34.44		
From time to time	81	22.82	24	13.33		
Regularly	13	3.66	0	0.00	4.64	0.001

Note: ^χ2^ presents variables where the differences were calculated by χ2. Missing values were not included in the analyses of the differences; the numbers in parentheses present the numerical values used for logistic regression analyses.

**Table 2 children-08-00877-t002:** Descriptive statistics (F—frequencies; %—percentages) for the study variables, with the differences between the groups according to the insufficiency/sufficiency of their physical activity levels (PAL) at the follow-up (MW—Mann–Whitney test; χ2—Chi square test).

	Insufficient PAL		Sufficient PAL		MW/χ^2^	
	F	%	F	%	Z/χ^2^	*p*
Gender ^χ2^						
Male	175	44.19	109	78.42		
Female	220	55.56	26	18.71		
Missing	1	0.25	4	2.88	53.78	0.001
Grade point average						
Excellent (5)	160	40.40	45	32.37		
Very good (4)	145	36.62	55	39.57		
Good (3)	73	18.43	31	22.30		
Sufficient (2)	4	1.01	3	2.16		
Insufficient (1)	13	3.28	4	2.88		
Missing	1	0.25	1	0.72	1.54	0.12
School absences						
<5 h (4)	169	42.68	60	43.17		
5–10 h (3)	158	39.90	50	35.97		
11–20 h (2)	58	14.65	22	15.83		
>20 h (1)	11	2.78	7	5.04		
Missing	0	0.00	0	0.00	0.35	0.72
Unexcused absences						
<5 h (5)	284	71.72	96	69.06		
6–10 h (4)	77	19.44	21	15.11		
11–15 h (3)	17	4.29	7	5.04		
16–20 h (2)	7	1.77	8	5.76		
>20 h (1)	11	2.78	7	5.04		
Missing	0	0.00	0	0.00	1.01	0.31
Behavioral grade						
Excellent (4)	343	86.62	113	81.29		
Very good (3)	37	9.34	15	10.79		
Proper (2)	9	2.27	7	5.04		
Poor (1)	7	1.77	4	2.88		
Missing	0	0.00	0	0.00	1.59	0.11
Sport participation						
Never been involved	128	32.32	17	12.23		
<1 year	83	20.96	24	17.27		
2–5 years	120	30.30	34	24.46		
>5 years	65	16.41	64	46.04		
Missing	0	0.00	0	0.00	6.67	0.001
Parental education						
Elementary	34	8.59	6	4.32		
High school	291	73.48	97	69.78		
College degree	43	10.86	17	12.23		
University degree	28	7.07	19	13.67		
Missing	0	0.00	0	0.00	2.55	0.01
Parental conflict						
Never	142	35.86	71	51.08		
Rarely	156	39.39	48	34.53		
From time to time	87	21.97	18	12.95		
Regularly	11	2.78	2	1.44		

Note: ^χ2^ presents variables where the differences were calculated by χ2. Missing values were not included in the analyses of differences; the numbers in parentheses present the numerical values used for the logistic regression analyses.

## Data Availability

The data is freely available here: http://www.kifst.hr/~dado/index_files/datacovid.sta (accessed on 30 September 2021).

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
