# Peer review of "The Impact of Scholastic Factors on Physical Activity Levels during the COVID-19 Lockdown: A Prospective Study on Adolescents from Bosnia and Herzegovina"

_children, 2021, doi:10.3390/children8100877_

Round 1

Reviewer 1 Report

Comments related to the manuscript ” The impact of scholastic factors on physical activity levels during the COVID-19 lockdown: A prospective study on adolescents from Bosnia and Herzegovina”

Major comments

In general:

What was the situation in Bosnia and Herzegovina with physical education? Did they cancel physical education totally, were there online lessons or assignments for students?

Minor Comments

Abstract:

Line 19-20        Why academic achievement are hypothesized to be important determinants of health-related behaviors? Usually socioeconomic background is more influential than personal academic achievement.

Line 21 Physical activity (PAL), why PAL, why not PA as usually?

Main content

Line 40 Now there is abbreviation PA?

Line 41 Maybe report something about the level of sufficient/insufficient physical activity in Bosnia and Herzegovina also in here?

Line 46-47        At this point there could be some references that highlights the importance of childhood and adolescence physical activity as a predictor of physical activity in adulthood and especially the tracking of low levels of physical activity and relation also to other health habits. E.g. Smoking and Physical Activity Trajectories from Childhood to Midlife https://www.mdpi.com/1660-4601/16/6/974

Line 54 In general, PA in adulthood…? I think you should clarify this as in the same sentence you are referring to previous PA. or if you are talking about adolescence, then mention previous PA in childhood.

Line 63 Previously PAL was physical activity, now PAL is physical activity levels, please be consistent.

Line 75-78        These studies are conducted in different kind of cultures. That may be one reason behind the results? Probably pointing this somehow? Or at least to mention that where these results come from?

Line 101           I’m a bit confused about reference [23]. That study did not measured physical literacy of children. However, in that manuscript reference [4] did so. I think that you should use reference |4] from that article instead of [23].

Line 114           Probably ad environmental crises (that are apart from weather crisis, e.g. earthquake, etc) here. These kinds of exceptional events may cut the possibility to participate to school and organized sports?

Line 147->        I wonder whether there is sense to analyze PAL with full package of PAQ-A before COVID-19 pandemic and during the Pandemic? IF there is physical education lessons and school recess in the PAQ-A questionnaire used, I think it will be clear that there will be difference. Should these both be analyzed without these two and consider, whether Covid has influenced to the time, that was not spent in school?

Line 264           Again I would consider to think about the difference in the cultures, e.g. in Korea there might be really hard pressures to gain good grades and these cultures are probably less concentrated on sports and physical activities while e.g. England and US are more concentrated on sports and organized sports + leisure time physical activity.

Line 269            How about another hobbies, like playing an instrument? Would it be possible that students spent their time to other hobbies than sports?

Line 305->        It should be noted that among adolescents, substance use (except tobacco use) participating to sports may not have protective effect. This is especially true among team sports where either drinking may be as usual as non-participants. And with snus use, it may be even more common among sport participants than nonparticipants.

Line 358           How about influence of GPA to education level? People with better GPA tend to go non-manual low and non-manual high works instead of manual work. Manual workers may have more exhaustive job or more unregular working times influencing to the possibility to leisure-time physical activity?

Line 369           I wonder, why in this paragraph there is no reference to the physical literacy as it was mentioned before. Higher physical literacy may promote physical activity as individuals have higher confidence and self-esteem to be physically active even though they are facing some new, maybe unfamiliar sports due to pandemic restrictions.

Line 384           physical activity -> PA

Line 425           PAQ-A does not measure the intensity of the respondent. It could be hypothesized that during the pandemic intensity of physical activity has declined (because of lack of trainings etc.). So, in addition to the decline of physical activity level also intensity might have declined. This could be mentioned as a limitation and probably discuss about the possible decline of intensity as well. Maybe find a reference that has already find this?

References

Line 531           There is some extra information “BMJ Open Sport & Exercise Medicine”

Line 552           index medicus ?

Line 558           index medicus ?

Line 575           Index Medicus ?

Line 578            index Medicus ?

Line 612           index medicus ?

Line 635           index medicus ?

Author Response

Comments related to the manuscript ” The impact of scholastic factors on physical activity levels during the COVID-19 lockdown: A prospective study on adolescents from Bosnia and Herzegovina”

Major comments

In general:

What was the situation in Bosnia and Herzegovina with physical education? Did they cancel physical education totally, were there online lessons or assignments for students?

RESPONSE: Thank you for this suggestion, it has been added in the text: “During the lockdown, schools were completely closed, and classes were held online. Regarding PE, it was also conducted online. There was no uniformed conduction of online PE, but each teacher made classes as they wanted and were able to, according to technical equipment and organizing skills. Mostly, PE was held in form of encouraging students to be active in their home or safe open spaces, with information about proper exercise form”. (Please see Methods section 2nd paragraph)

Minor Comments

Abstract:

Line 19-20        Why academic achievement are hypothesized to be important determinants of health-related behaviors? Usually socioeconomic background is more influential than personal academic achievement.

RESPONSE: Thank you for this suggestion. The explanation has been added in the last paragraph of the Introduction. Text reads: “We hypothesized that students with better academic achievement are generally acting more responsible towards themselves and their success. Also, we theorised that they will act more responsible towards their health and will have the knowledge and awareness about importance of PA for their health, and that scholastic factors would be protective against PA decline in adolescents.”. (please see last part of the Introduction)

We agree that socioeconomic background is also very influential, hence, that is the reason why we included parental education, parental conflict, sport participation, age and gender as potential confounders. However, in the previous studies we did not find that SES is associated with physical activity and therefore we did not include it as a confounder in this study. Please see: https://pubmed.ncbi.nlm.nih.gov/33673435/

Line 21 Physical activity (PAL), why PAL, why not PA as usually?

RESPONSE: Thank you for noticing this. This was a typing error. We are using both PA and PAL abbreviation throughout the manuscript. PA stands for physical activity and PAL stands for physical activity levels.

Main content

Line 40 Now there is abbreviation PA?

RESPONSE: Thank you again for noticing it. We corrected abbreviations and hope now is clear.

Line 41 Maybe report something about the level of sufficient/insufficient physical activity in Bosnia and Herzegovina also in here?

RESPONSE: Thank you for this suggestion. We added the information about physical activity in Bosnia and Herzegovina. Text reads:Supportively, a recent paper reported that 56% of girls and 44% of boys in Bosnia and Herzegovina have insufficient PA [3]”.  Please see 1st paragraph of the Introduction.

Line 46-47        At this point there could be some references that highlights the importance of childhood and adolescence physical activity as a predictor of physical activity in adulthood and especially the tracking of low levels of physical activity and relation also to other health habits. E.g. Smoking and Physical Activity Trajectories from Childhood to Midlife https://www.mdpi.com/1660-4601/16/6/974

RESPONSE: Thank you for this suggestion. We also consider this as very important to point out. We added the phenomenon of the tracking of physical activity. Text now reads: “What is more, there are firm evidences that health habits during adolescence track into adulthood, with low PA tracking better than high PA [5]. Finnish study found that adolescents with low PA had higher probability of being smokers in adulthood, which additionally detriments health [6]”. Please see first paragraph of the Introduction

Line 54 In general, PA in adulthood…? I think you should clarify this as in the same sentence you are referring to previous PA. or if you are talking about adolescence, then mention previous PA in childhood.

RESPONSE: Thank you for this comment and suggestion. Indeed, the sentence and reference were referring to general population (from children to adults). Therefore, we removed that reference and used more appropriate one: Park, H.; Kim, N. Predicting factors of physical activity in adolescents: a systematic review. Asian Nurs Res (Korean Soc Nurs Sci) 2008, 2, 113-128, doi:10.1016/s1976-1317(08)60035-3.
The amended text now reads: “In general, adolescents’ PA is positively associated with male gender, perceived activity competence, intentions to exercise, previous childhood PA, social support, family cohesion, families’ high socioeconomic status, and opportunities to exercise [12]”. (please see 2nd paragraph of the Introduction)

Line 63 Previously PAL was physical activity, now PAL is physical activity levels, please be consistent.

RESPONSE: Thank you for noticing this inconsistency. We changed the abbreviations and used them throughout the manuscript (PA – physical activity; PAL – physical activity level). We hope the issue is clear now.

Line 75-78        These studies are conducted in different kind of cultures. That may be one reason behind the results? Probably pointing this somehow? Or at least to mention that where these results come from?

RESPONSE: Thank you for this valuable comment. We agree that cultural differences probably/surely exist. We amended this paragraph, text now reads: “However, the associations between academic achievement and PAL are not conclusive; study on Chinese adolescents has evidenced no association [18] while a study on Korean adolescents evidenced a negative association [19] between these factors. Therefore, the problem should be further examined, and special emphasis should be placed on possible cultural differences that may have caused such inconclusive results”. (please see end of the 4th paragraph of the Introduction)

Line 101           I’m a bit confused about reference [23]. That study did not measured physical literacy of children. However, in that manuscript reference [4] did so. I think that you should use reference |4] from that article instead of [23].

RESPONSE: We agree, and thus we used the suggested reference instead.

Line 114           Probably ad environmental crises (that are apart from weather crisis, e.g. earthquake, etc) here. These kinds of exceptional events may cut the possibility to participate to school and organized sports?

RESPONSE: Thank you for this suggestion. Environmental crises definitely have to be mentioned as they will surely limit the possibility of participating in regular life, such as going to school or practicing sports. Therefore, we added it in the text, and now it reads: “We were of the opinion that the results of this study would be applicable to the development of targeted campaigns against the known and widespread decrease of PAL in adolescence, not only in the situation of the COVID-19 pandemic, but also during other health crises, weather crises, environmental crises, or other situations where there may be limited movement opportunities”. Please see the last paragraph of the Introduction.

Line 147->        I wonder whether there is sense to analyze PAL with full package of PAQ-A before COVID-19 pandemic and during the Pandemic? IF there is physical education lessons and school recess in the PAQ-A questionnaire used, I think it will be clear that there will be difference. Should these both be analyzed without these two and consider, whether Covid has influenced to the time, that was not spent in school?

RESPONSE: Thank you for this comment. We agree that PAQ-A refers to school days that include live PE classes. However, as this study started before the pandemic we used the full PAQ-A, and did the same even during the pandemic to preserve consistency. However, we agree that some questions are not applicable for the lockdown period. This is now added in the Limitations section: “Also, this study used the complete PAQ-A questionnaire, with the possibility that some questions are not applicable for the lockdown period. However, we did include the full questionnaire as we initially collected all data and continued the same during the lockdown”.

Line 264           Again I would consider to think about the difference in the cultures, e.g. in Korea there might be really hard pressures to gain good grades and these cultures are probably less concentrated on sports and physical activities while e.g. England and US are more concentrated on sports and organized sports + leisure time physical activity.

RESPONSE: Thank you for this comment. We agree that there exist cultural differences regarding this issue. However, we did not want to detailly point this out in order to keep readers attention and more focus on obtained results. However, we did point this out, now text reads: “There could be several possible explanations such as cultural differences, environmental situations, but the authors of that study explained such findings by the type of school”. (please see subheading 4.1)

Line 269            How about another hobbies, like playing an instrument? Would it be possible that students spent their time to other hobbies than sports?

RESPONSE: Yes, definitely. Thank you for this comment. This is also the possibility but in this paper and study we wanted to keep the focus on physical activity.

Line 305->        It should be noted that among adolescents, substance use (except tobacco use) participating to sports may not have protective effect. This is especially true among team sports where either drinking may be as usual as non-participants. And with snus use, it may be even more common among sport participants than nonparticipants.

 RESPONSE: We completely agree. Indeed, we published several papers regarding this topic. Briefly, adolescents involved in sports are more likely to drink alcohol and less likely to smoke than their peers that are not involved in sports, while snus is not used in Southeastern Europe and we do not have information about it. However, even though this is a very interesting and important issue, we did not want to go in more detail in this manuscript as we consider that it would distract readers attention. But we did additionally accentuate that smoking is mainly associated with low physical activity and not participating in sports. Text reads: “Therefore, children with poor behavior grades are often those children who partake in health-risk behavior (such as SUM, especially smoking)”. Please see end of the 2nd paragraph of the subheading 4.2

Line 358           How about influence of GPA to education level? People with better GPA tend to go non-manual low and non-manual high works instead of manual work. Manual workers may have more exhaustive job or more unregular working times influencing to the possibility to leisure-time physical activity?

 RESPONSE: This could definitely be the case, but we investigated adolescents who are not working yet; thus, adding this in the manuscript is not possible. However, we will have this in mind for future investigations, thank you.

Line 369           I wonder, why in this paragraph there is no reference to the physical literacy as it was mentioned before. Higher physical literacy may promote physical activity as individuals have higher confidence and self-esteem to be physically active even though they are facing some new, maybe unfamiliar sports due to pandemic restrictions.

 RESPONSE: Thank you for pointing this out. We added it in the manuscript, text now reads: “Also, those students might possess better physical literacy, which gave them confidence and self-esteem for being physically active even in different space and limited equipment for practicing sports or exercising [52]”. (please see 3rd paragraph of the subheading 4.4).

Line 384           physical activity -> PA

 RESPONSE: Thank you for noticing this, amended accordingly.

Line 425           PAQ-A does not measure the intensity of the respondent. It could be hypothesized that during the pandemic intensity of physical activity has declined (because of lack of trainings etc.). So, in addition to the decline of physical activity level also intensity might have declined. This could be mentioned as a limitation and probably discuss about the possible decline of intensity as well. Maybe find a reference that has already find this?

RESPONSE: This is absolutely true, thank you. We added it to the limitations section and added a study conducted on Croatian adolescents which included the intensity. (Karuc, J.; Soric, M.; Radman, I.; Misigoj-Durakovic, M. Moderators of Change in Physical Activity Levels during Restrictions Due to COVID-19 Pandemic in Young Urban Adults. Sustainability 2020, 12, 6392, doi:10.3390/su12166392.)

References

Line 531           There is some extra information “BMJ Open Sport & Exercise Medicine”

Line 552           index medicus ?

Line 558           index medicus ?

Line 575           Index Medicus ?

Line 578            index Medicus ?

Line 612           index medicus ?

Line 635           index medicus ?

RESPONSE: Thank you for noticing this, we amended those references. Please see References section.

Staying at your disposal

Authors

Reviewer 2 Report

The first question for the authors refers to the fact that there is an explanation for which only girls were evaluated in this study .... why boys were not chosen? Please see lines 23,24....and ''46% females''....and the rest, i.e 54% are boys? I see you have:

Male

284

53.1

Please correct on paper....it is important to understand exactly all the information

It would be very interesting for the authors to add to the chapter Introduction and what are the WHO recommendations regarding the level of physical activity for adolescents....I don't see in this chapter and it is important for this paper.

The authors say in abstract: ''The participants were adolescents form Bosnia and Herzegovina (n = 525, 46% females), .....and in Methods: " This prospective study included 721 adolescents from Bosnia and Herzegovina (43% female).....what is the exact number of participants? Was 721 teenagers in the initial phase and 525 in the end? please explain very clearly the methodology of recruiting participants

Line 134 - please put point after title of figure.

It is necessary to enlarge the characters in the figure, the writing is not very clearly understood.

line 143 - to much space between words, see ''Adolescents (PAQ-A). Adolescents filled''...please correct. Also, line 150 etc...

PAQ-A - by whom this questionnaire was validated and what was the alpha value of this assessment tool

Also present the framing values ​​of the questionnaire, not only that ''The results of each item and the total score are scaled from 1 to 5, representing low too high''....I.E:

1. LOW PAL - VALUES .........
2. AVERAGE PAL .... VALUES .......
3. HIGH PAL - VALUES........

It is too little to say sufficient or insufficient values, we must see very clearly the values ​​for each participant or their average of physical activity, both before and during e lockdown period.

As I said before, you can make a classification of the type of physical activity: low, medium, high ...... in Table 1, not only enough and insufficient. Please bold and punctuate the table title, ALSO FOR TABLE 2

Author Response

The first question for the authors refers to the fact that there is an explanation for which only girls were evaluated in this study .... why boys were not chosen? Please see lines 23,24....and ''46% females''....and the rest, i.e 54% are boys? I see you have:

Male

284

53.1

Please correct on paper....it is important to understand exactly all the information

RESPONSE: Thank you for this comment and suggestion. The inconsistency in “numbers” occurred as a result of some participants not reporting their gender. Because of that, in the Table 1 we included the option “Missing” for reporting Gender.

It would be very interesting for the authors to add to the chapter Introduction and what are the WHO recommendations regarding the level of physical activity for adolescents....I don't see in this chapter and it is important for this paper.

RESPONSE: Thank you for this suggestion. We agree that this information is important, thus it stands in the Introduction section. Text reads: “Specifically, 81% of adolescents are reported to be insufficiently physically active worldwide, meaning that they do not reach the WHO’s recommendation of 60 minutes of PA a day”.

The authors say in abstract: ''The participants were adolescents form Bosnia and Herzegovina (n = 525, 46% females), .....and in Methods: " This prospective study included 721 adolescents from Bosnia and Herzegovina (43% female).....what is the exact number of participants? Was 721 teenagers in the initial phase and 525 in the end? please explain very clearly the methodology of recruiting participants

RESPONSE: Thank you for this comment. Indeed, we had 721 participants at study baseline, and due to drop out of some participants the total number was 525. We corrected and clarified it in the text now. Please see 1st paragraph Methods section. Text reads: “This prospective study included total of 525 adolescents from Bosnia and Herzegovina (43% female)”.

Line 134 - please put point after title of figure.

RESPONSE: It is now corrected, thank you.

It is necessary to enlarge the characters in the figure, the writing is not very clearly understood.

RESPONSE: Thank you for this suggestion. The figure has been amended accordingly.

line 143 - to much space between words, see ''Adolescents (PAQ-A). Adolescents filled''...please correct. Also, line 150 etc...

RESPONSE: Thank you for noticing this, it is now corrected.

PAQ-A - by whom this questionnaire was validated and what was the alpha value of this assessment tool

RESPONSE: PAQ-A for Bosnia and Herzegovina and Croatia was validated in the study by Vidranski et al. (2020) Alpha value was 0.87. Please see: http://actakinesiologica.com/wp-content/uploads/2020/12/10-Vidranski-T.-et.-al..pdf

Also present the framing values ​​of the questionnaire, not only that ''The results of each item and the total score are scaled from 1 to 5, representing low too high''....I.E:
1. LOW PAL - VALUES .........
2. AVERAGE PAL .... VALUES .......
3. HIGH PAL - VALUES........

It is too little to say sufficient or insufficient values, we must see very clearly the values ​​for each participant or their average of physical activity, both before and during e lockdown period.

As I said before, you can make a classification of the type of physical activity: low, medium, high ...... in Table 1, not only enough and insufficient. Please bold and punctuate the table title, ALSO FOR TABLE 2

RESPONSE: Yes, indeed, your suggestion is absolutely reasonable. It is certain that some participants were clustered into separate groups but still had very similar PAL as you said. However, the division into three groups will make statistical analyses and interpretations more complicated (we should use multinomial and not logistic regression). At the same time, we obtained relatively logical and interpretable results using the “two-group clustering”, so for a moment we will rather hold our original approach. Of course, if you will insist on changes and three-group-clustering, we will certainly follow your advice. For a moment, the problem is in this version of the paper specified in the Limitations section. Text reads: “In this research results on PAQ-A were clustered into two groups, which undoubtedly allocated participants with similar results into opposed groups. In the future studies, division into three groups is therefore suggested”. (please see subsection Limitations and Strengths, highlighted text).

Furthermore, the information about PAL at baseline and follow-up is presented in the first part of the Results section. Text reads: “The PAL significantly declined between baseline and follow-up (2.43±0.71 and 2.00±0.75, respectively; t-test = 4.14, p < 0.001), indicating negative impact of the COVID-19 imposed lockdown on PAL among studied adolescents”.

Staying at your disposal

Authors

Round 2

Reviewer 1 Report

Thank you for the improvements, I think it has improved. 

Reviewer 2 Report

Dear authors,

Congrats on all changes in your manuscript. It is clearer than previous version.